# Genome-Assisted Probiotic Characterization and Application of *Lactiplantibacillus plantarum* 18 as a Candidate Probiotic for Laying Hen Production

**DOI:** 10.3390/microorganisms11102373

**Published:** 2023-09-22

**Authors:** Guoqing Zhang, Ning Yang, Zhongyuan Liu, Xinyu Chen, Mengjiao Li, Tongyu Fu, Donghong Zhang, Cuiqing Zhao

**Affiliations:** College of Animal Science and Technology, Jilin Agricultural Science and Technology University, Jilin 132101, China; zguoqing9908@163.com (G.Z.);

**Keywords:** *Lactiplantibacillus plantarum*, genome, laying hen, productivity

## Abstract

Probiotics gained significant attention for their potential to improve gut health and enhance productivity in animals, including poultry. This comprehensive study focused on the genetic analysis of *Lactiplantibacillus plantarum* 18 (LP18) to understand its survival and colonization characteristics in the gastrointestinal tract. LP18 was supplemented in the late-stage diet of laying hens to investigate its impact on growth performance, egg quality, and lipid metabolism. The complete genome sequence of LP18 was determined, consisting of 3,275,044 base pairs with a GC content of 44.42% and two circular plasmids. Genomic analysis revealed genes associated with adaptability, adhesion, and gastrointestinal safety. LP18 supplementation significantly improved the daily laying rate (*p* < 0.05) during the late-production phase and showed noteworthy advancements in egg quality, including egg shape index (*p* < 0.05), egg albumen height (*p* < 0.01), Haugh unit (*p* < 0.01), and eggshell strength (*p* < 0.05), with notable improvements in eggshell ultrastructure. Additionally, LP18 supplementation resulted in a significant reduction in serum lipid content, including LDL (*p* < 0.01), FFA (*p* < 0.05), and Gly (*p* < 0.05). These findings provide valuable insights into the genomic characteristics of LP18 and the genes that support its survival and colonization in the gastrointestinal tract. Importantly, this study highlights the potential of LP18 as a probiotic candidate to enhance productivity, optimize egg quality, and modulate lipid metabolism in poultry production.

## 1. Introduction

Probiotics are live microorganisms that have health benefits when consumed in sufficient amounts [1] and are mainly found in the intestinal tract of the body. Probiotics have a long history of safe use in various fields, such as food processing, agriculture, livestock farming, and health care. Probiotics can provide several health benefits to the host by interfering with potential pathogens, improving barrier function, immunomodulation, and neurotransmitter production [2]. Numerous probiotics are employed in the poultry industry to optimize animal health and enhance egg quality while minimizing the cost of feed. Previous research indicates that the inclusion of probiotics in the diet of poultry showed the potential to improve feed conversion efficiency, enhance hen performance, and enhance egg quality [3,4,5]. Furthermore, probiotic dietary supplementation was observed to promote serum immune responses and antioxidant function [6], as well as contribute to the maintenance of intestinal health in laying hens [7,8]. However, it is worth noting that the efficacy of probiotics is strain- and disease-specific [9]. The Food and Drug Administration (FDA) in the United States designated the genus *Lactobacillus* as generally recognized as safe (GRAS), and the European Food Safety Authority (EFSA) listed it as having a qualified presumption of safety (QPS), which guarantees its use in food and safety in humans, respectively. *Lactiplantibacillus plantarum* (*L. plantarum*) is one of the largest known genomes of probiotics and belongs to the genus *Lactobacillus* [10]. *L. plantarum* can be found adhering and colonizing the intestinal tract [11,12] and beneficially modulating the immune system by producing cytokines and enhancing phagocytic cells’ activity. Among the extensively utilized probiotics, *Lactiplantibacillus* strains were shown to stimulate growth performance, improve meat quality, enhance immune response, and prevent certain avian diseases [13]. *L. plantarum* was demonstrated to enhance egg production and feed intake in laying hens, thereby influencing the composition of fecal microbiota and improving intestinal development and digestive capacity [14].

Among these strains, *L. plantarum* 18 (LP18) emerged as a highly promising probiotic candidate, exhibiting strong adherence to small intestinal cells [15], anti-inflammatory and immunomodulatory properties [16], and the ability to prevent inflammatory disorders caused by intestinal pathogens [17], while showing outstanding probiotic properties and safety in vitro [18]. To gain insight into the probiotic characteristics of LP18 and its potential application in laying hen production, in this study, we comprehensively analyzed the LP18 whole genome sequence. Additionally, the efficacy of LP18 on the performance and egg quality of laying hens was also evaluated in the late production stage.

## 2. Materials and Methods

### 2.1. Bacterial Growth and DNA Extraction

The LP18 (CGMCC 1.557 NZ_CP016270.1) strain was procured from the China General Microbiological Culture Collection (CGMCC) and was stored in MRS medium supplemented with 25% (*v*/*v*) glycerol as a cryoprotectant at −80 °C. Subsequently, LP18 was inoculated in MRS medium at a volume of 1% (*v*/*v*) and grown for 24 h at 37 °C. Following this, the organisms were collected by centrifugation at 8000× *g* for 5 min.

Genomic DNA extraction was carried out using the SDS method, as described previously [19]. The resultant DNA sample was assessed by agarose gel electrophoresis to verify its quality, while the purity and concentration of the total genomic DNA were determined by measuring the absorbance.

### 2.2. Genome Sequencing, Assembly

The 1 µg amount of genomic DNA per sample was used as an input material and fragmented by sonication to a size of 350 bp. The fragmented DNA was subjected to end polishing, A-tailing, and ligation with a full-length adaptor for Illumina sequencing, followed by PCR amplification. The PCR products were then purified using the AMPure XP system (Beckman Coulter, Brea, CA, USA), and libraries were analyzed for size distribution by the Agilent 2100 Bioanalyzer (Agilent, Santa Clara, CA, USA) and quantified using real-time PCR. The Illumina NovaSeq PE150 technology provided by Beijing Novogene Bioinformatics Technology Co., Ltd. (Beijing, China) was applied for sequencing of the whole genome of LP18. Raw data obtained from sequencing had a proportion of low-quality data, which could compromise the accuracy and the reliability of subsequent information analysis results if left unfiltered. Therefore, the raw data underwent filtering to obtain valid data (clean data) before proceeding with genome assembly. The genomic map of LP18 was generated using CGView.

### 2.3. Genome Annotation

Protein coding sequences (CDS) and functional annotation of predicted genes were performed using the RAST online server (http://www.rast.nmpdr.org/, accessed on 15 July 2023) to identify tRNA and rRNA sequences. Gene function prediction was accomplished by employing eggNOG 5.0 for gene file annotation [20], generating files with annotations such as gene ontology (GO), Kyoto Encyclopedia of Genes and Genomes (KEGG), clusters of orthologous groups (COG), enzyme commission (EC), and Carbohydrate-Active Enzymes database (CAZy).

### 2.4. Phylogenetic Tree Construction

A comparative genomic analysis was conducted on LP18 to confirm its genetic characteristics, using 18 probiotic Lactiplantibacillus strains obtained from the National Center for Biotechnology Information (NCBI) genome database. In order to estimate the phylogenetic tree, 16S rRNA gene sequences from the genome sequence of 19 probiotic Lactiplantibacillus strains were aligned through ClustalW with default parameters and then subjected to phylogenetic analysis using the maximum-likelihood method in MEGA (version 11).

### 2.5. Safety Assessment

#### 2.5.1. Hemolysis Assay

A sterile inoculation loop with a volume of 1 μL was employed to streak the activated probiotic bacterial suspension onto Columbia blood agar plates. The plates were then incubated at 37 °C for 48 h, and the presence of a hemolytic zone around the colonies was observed. Beta-hemolytic Staphylococcus aureus subsp. aureus Rosenbach (*S. aureus*, BNCC310011, BeNa Culture Collection, Beijing, China) and gamma-hemolytic *Lactobacillus rhamnosus* GG (LGG, #53103, ATCC, St. Cloud, MN, USA) were used as controls.

#### 2.5.2. Acute Oral Toxicity Test

Forty C57BL/6 mice aged between 8 and 10 weeks, comprising 20 females and 20 males, were selected for the experiment. The mice were acclimatized for 5 days before the test. Subsequently, the male and female mice were divided into two groups, with each group consisting of 10 males and 10 females. The experimental group received a daily oral gavage of LP18 at a dose of 10^11^ colony-forming units (CFU/d), while the control group received PBS. This administration continued for 14 consecutive days. The behavior, mortality, signs of toxicity, and body weight of the mice were recorded daily.

#### 2.5.3. Genomic Security Assessment

In order to comprehensively assess the safety of LP18 from a genomic perspective, an analysis was conducted utilizing established databases, namely Virulence Factors of Pathogenic Bacteria (VFDB). This database was utilized to evaluate the presence of virulence factors and antibiotic resistance genes in the genomic sequence of LP18. This approach enabled a thorough investigation into the potential risks associated with this bacterium and provided valuable insights into its overall safety profile.

### 2.6. Animal Trials

#### 2.6.1. Experimental Design

Healthy Jinghong No.1 laying hens (63 weeks old) exhibiting uniform egg production were selected and fed without antibiotics. They were randomly divided into CON (*n* = 30) and LP18 (*n* = 30) birds, with five replicate groups of each (six hens/group). Hens were fed in three-layer ladder cages with two hens per cage. The experimental trial period lasted for 45 days, including early (1–15 d), middle (16–30 d), and late (31–45 d) stages. During the trial, CON hens were fed a basal diet, and those in the LP18 group were fed the basal diet supplemented with 8 × 10^9^ cfu/kg LP18. The composition and nutritional values of the basal diet are shown in Appendix A.

The experimental protocols used in this experiment, including animal care and use, were reviewed and approved by the Animal Care and Use Ethics Committee of Jilin Agricultural Science and Technology University (Jilin, China) (LL2021017).

#### 2.6.2. Production Performance

The feed intake, egg yield, and egg weight of all hens were recorded daily throughout the trial. The feed conversion rate and laying rate were calculated for each trial stage. The laying rate was calculated as the rate of egg production (including normal and broken eggs) per hen per day.
Feed conversion ratio = Feed consumption (kg)/total egg weight (kg).

#### 2.6.3. Egg Quality

At the end of the experimental period (45 d), 20 to 30 eggs were randomly selected from each group and their quality was determined. The weight (w) of each egg was recorded and its length and width were measured using an egg shape tester (FHK, Tokyo, Japan). Eggshell strength was measured using an eggshell strength tester (FHK, Japan). The eggshell thickness was measured at three locations (sharp end, equator, and air cell) using an eggshell thickness tester (FHK, Japan), and the mean was calculated. An egg was broken onto a glass plate, the diameter of the yolk was measured using vernier calipers, and its color was identified using a yolk color chart. The height of the egg white and yolk was measured at three locations using an egg white height tester (FHK, Japan). Based on these data, the egg shape index, yolk index, and Haugh unit were calculated. Egg shape index (%) = (egg width/egg length) × 100; Haugh units = 100 × log (H − 1.7 W^0.37^ + 7.57); and yolk index (%) = (yolk height/yolk diameter) × 100.

#### 2.6.4. Eggshell Ultrastructure

At the end of the trial, three eggs were randomly selected from each group, and the white and yolk were removed. A sample of eggshell (approximately 1 cm^2^) was cut from the three locations and the shell membrane was peeled off (inwards from the edge) using forceps. Eggshell samples were boiled for 10 min in 2% NaOH solution and air dried at room temperature for 1 d. Samples were then stuck to the sample stage of a Hitachi ion-sputtering instrument using double-sided adhesive, and electrically conductive carbon tape and sprayed for approximately 30 s. The cross-section and inner and outer surfaces of the eggshell were observed under a scanning electron microscope (Hitachi, Japan). The mammillary thickness, effective layer thickness (total palisade, vertical crystal layer, and cuticle thicknesses), and width of the mastoid knot were measured.

#### 2.6.5. Serum Biochemical Indicators

At the end of the trial, blood was drawn from hens in each group, left to stand at room temperature for 2 h, and centrifuged at 2500 rpm and 4 °C for 30 min. The concentrations of triglyceride, low-density lipoprotein, non-esterified fatty acids, glycerol, and total cholesterol were measured in the blood serum in accordance with test kit instructions (Nanjing Jiancheng Institute of Bioengineering, Nanjing, Jiangsu, China).

### 2.7. Statistical Analyses

Excel and GraphPad Prism 7 were used for statistical analyses. Results are expressed as mean ± standard error of the mean, and group data were compared by *t*-test. Differences were considered significant, very significant, and extremely significant when * *p* < 0.05, ** *p* < 0.01, and *** *p* < 0.001, respectively.

## 3. Results

### 3.1. General Genome Features of LP18

LP18, a prokaryotic organism characterized by circular chromatin, was the subject of thorough genomic analysis in this study. Via advanced sequencing technology, the sample LP18 genome was successfully assembled with a total length of 3,275,044 bp and a GC content of 44.42% (Table 1). Comprehensive genomic analysis revealed that the LP18 genome contained 3142 genes totaling 2,732,562 bp in length, with an average length of 870 bp and accounting for 83.44% of the total genome length. Additionally, 86 tandem repeat sequences with a total length of 16,584 bp were identified, which accounted for 0.51% of the total genome length. The presence of 48 minisatellite sequences and 4 microsatellite sequences, as well as 68 tRNAs and 13 rRNAs, add important insights into the genetic landscape of LP18. Furthermore, two distinct circular plasmid DNAs, namely pL1 and pL2, were discovered, with lengths of 67,395 bp and 49,005 bp, respectively. The GC content of pL1 and pL2 was measured at 40.2% and 39.1%, respectively. Finally, an analysis of GC bias in sequencing using GC depth demonstrated that the points of each sequence were concentrated around 44% GC content, indicating the high quality of the sequencing results (Figure 1). The sequencing data were deposited in the NCBI database under the accession numbers: chromosomal sequence (NZ_CP016270.1); plasmid pL1 (NZ_CP016271); and plasmid pL2 (NZ_CP016272.1).

In the LP18 genome, we identified a CRISPR locus measuring 564 bp, housing a conserved direct repeat sequence (GTCTTGAATAGTAGTCATATCAAACAGGTTTAGAAC) and encompassing eight distinct spacers. Remarkably, the CRISPR/Cas system discovered in the LP18 genome conforms to the archetypal type II CRISPR system. Notably, this system presents the presence of the Cas9 protein alongside Cas1 and Cas2 components, while lacking a Csn2-related protein. Moreover, it is noteworthy that the endonuclease activity is exclusively exhibited by the Cas9 protein (Appendix A).

Furthermore, the genome of LP18 comprises three genes involved in the production of plantaricin, including a regulatory operon *plnD*, a two-peptide bacteriocin EF encoded by *plnF*, and a transport operon *plnU* (Appendix A).

### 3.2. Genome Functional Annotation Analysis of LP18

RAST was employed to assign the annotated genes to their corresponding subsystems, as depicted in Figure 2A. Within the LP18 genome, a predominant proportion of the annotated genes were implicated in carbohydrate metabolism, succeeded by amino acid and protein metabolism, while no nitrogen fixation-associated genes were identified. To comprehend the functional allocation of LP18 genes within metabolic pathways and cellular processes, KEGG annotation (Figure 2B) unveiled that LP18 genes were mainly enriched in membrane transport (416) within the environmental information processing category, as well as in carbohydrate metabolism (327) and amino acid metabolism (233) within the metabolism category. For investigating the functionality, interrelation, and precise interpretation of gene expression and regulation, GO annotation (Figure 2C) indicated significant enrichment in the metabolism process (1098), catalytic activity (1089), and cellular process (1014). In the CAZy database (Figure 2D), LP18 genes were primarily enriched in glycosyl transferases (GTs) and glycoside hydrolases (GHs), suggesting the presence of potential pathways for carbohydrate synthesis and degradation. In terms of gene participation in essential biological processes within cells, according to the COG database (Figure 2E), among the 19 functional categories, 23% (579) of genes had unknown functions, 272 genes were responsible for transcription, and 217 genes were involved in replication, recombination, and repair. To evaluate the phylogenetic distance among different Lactiplantibacillus species, including LP18, a phylogenetic tree based on 16S rRNA was constructed (Figure 2F), encompassing 19 Lactiplantibacillus strains. Within the Lactiplantibacillus genus, LP18 exhibited the closest relation to *L. plantarum* SBM 42968 (100% identical), followed by *L. plantarum* GB 348 and *L. plantarum* NBRC 15891.

### 3.3. Analysis of LP18 Probiotic Properties Based on Annotated Information

#### 3.3.1. Gastrointestinal Tract Survival Potential of LP18

Gastrointestinal tolerance is a crucial trait required by LP18, encompassing variations in temperature, pH, and salinity. As shown in Figure 2, the LP18 genome harbors a total of two genes encoding the F0F1-ATPase, the primary intracellular pH regulatory factor. Further, nine genes are identified encoding sodium proton antiporter/Na-H antiporters responsible for maintaining homeostasis between Na^+^ and H^+^. In addition, two genes encoding alkaline phosphatases were identified, along with two genes encoding *ArgR* proteins, which are associated with enhanced acid tolerance in the strain. Cold shock proteins (*CSPs*) are part of the stress adaptation system in *L. plantarum* strains, including LP18. Three cold shock protein genes, *cspC*, *cspP*, and *cspL*, were identified in the LP18 genome. Furthermore, a DEAD-box RNA helicase (*cshB*) was found to work in conjunction with the cold shock proteins to ensure normal transcription initiation at both low and optimal temperatures. The presence of *HSP1*, *HSP2*, and *HSP3*, encoding members of the small heat shock protein 20 (*HSP20*) family, was identified in the LP18 genome. Specifically, *CspC* contributes to cell recovery from heat-induced damage, in collaboration with *DnaK*, *DnaJ*, and *GrpE*. Moreover, the negative regulator *HrcA* controls the first class of heat shock genes, thereby preventing their induction during heat shock [21]. Additionally, the LP18 genome encodes 26 ABC transport proteins and 2 ATP-binding proteins associated with osmotic stress.

Many stress-related proteins contribute to the adaptive responses of *L. plantarum* to gastrointestinal stress. The LP18 genome encodes chaperone proteins including *GrpE*, *DnaK*, and *DnaJ*, which form part of a stress-induced multi-chaperone system. The presence of the universal stress protein *UspA* and the NADH-dependent oxidoreductase gene (*dhaT*) in LP18 is crucial for bacterial oxidative stress response [22]. Additionally, key proteins involved in bile salt stress and adaptation are found in the LP18 genome, including glutathione reductase, which participates in the protection against oxidative damage caused by bile salts, as well as choloylglycine hydrolase, ABC transporters, and F0F1-ATPase, contributing to active clearance of bile-related stress factors [23]. The LP18 genome sequence also encompasses the *katA* gene, encoding heme-dependent catalase, which protects cells against oxidative stress [24]. Furthermore, the LP18 genome harbors proteases involved in stress response, including ATP-dependent intracellular proteases such as *ClpC*, *ClpE*, *ClpB*, *ClpX*, *ClpL*, *ClpP*, *Lon*, *HslU*, *HslV*, and *HslO*, which play vital roles in degrading non-functional and aberrant proteins, thus contributing to the defense system against oxidative stress (Table 2).

#### 3.3.2. Gastrointestinal Tract Colonization Potential of LP18

The LP18 genome encompasses a total of 16 genes encoding cell surface proteins, including 5 genes encoding cell surface protein, 1 gene encoding host cell surface-exposed lipoprotein, and 9 genes encoding WxL domain surface cell wall-binding proteins. These cell surface proteins were demonstrated to play roles in adhesion or binding to other cells. Notably, these include one *srtA* protein, one fibronectin-binding protein, ten mucus-binding proteins, five β-galactosidase, and glyceraldehyde-3-phosphate. Furthermore, the LP18 genome includes a gene encoding UDP-galactopyranose mutase, as well as 38 genes encoding glycosyltransferases, all of which facilitate bacterial–host interactions and contribute to the adhesion of bacterial cells to intestinal epithelial cells. Moreover, genes encoding ABC transporters and 11 proteins involved in the PTS system were identified in the genome, with their expression induced by mucin, thus enabling the establishment of bacterial colonization within the intestinal tract (Table 3).

### 3.4. Security Assessment of LP18

#### 3.4.1. Genomic Security Assessment and Hemolytic Activity of LP18

The genomic security assessment of *Lactiplantibacillus plantarum*, specifically LP18, revealed interesting results concerning potential antibiotic resistance and virulence-associated factors. The presence of the *vanY* gene within the *vanB* cluster suggests the potential for resistance to vancomycin, a glycopeptide antibiotic. Additionally, the *qacJ* gene was detected, indicating a potential mechanism for resistance to disinfecting agents and antiseptics, specifically benzalkonium chloride (Appendix A). The identification of various virulence-associated factors in the LP18 genome is notable. These factors include adherence genes, enzyme genes, immune evasion genes, iron acquisition genes, regulation genes, secretion system genes, toxin genes, and genes associated with other essential functions, such as antiphagocytosis and bile resistance (Appendix A).

The results presented in this study provide valuable insights into the security assessment of LP18 from multiple perspectives. As shown in Figure 3, the hemolysis test results demonstrate that LP18 does not exhibit β-hemolytic activity, contrasting with the positive control *S. aureus* that displayed a distinct transparent zone indicative of β-hemolysis. This observation suggests that LP18 is non-hemolytic, which is a desirable characteristic for probiotic strains.

#### 3.4.2. Acute Oral Toxicity Study of LP18

In the acute oral toxicity test, the administration of LP18 at a relatively high dose over a 14-day period did not result in significant differences in the average body weight of the treated mice compared to the control group (Table 4). Furthermore, there were no instances of mortality, toxic symptoms, or pathological changes observed in either group. These findings indicate the safety of LP18 at the tested dosage and duration of administration, supporting its potential application as a probiotic.

### 3.5. Effects of LP18 Supplementation on Late-Stage Layer Production

#### 3.5.1. Growth Performance

The current study investigates the effects of LP18 supplementation on egg production in laying hens. Table 5 summarizes the productive performance of laying hens. This includes data related to egg weight, feed intake, egg production rate, and feed–egg ratio. The addition of LP18 did not significantly affect the average egg production rate and feed–egg ratio of the hens during the early and mid-term phases of the experiment (*p* > 0.05). However, during the latter phase, compared with the CON group, the LP18 group showed a significant increase in the average egg production rate (7.9%, *p* < 0.01) and a significant decrease in a feed–egg ratio (8.1%, *p* < 0.05). LP18 did not have a significant effect on the average egg weight at any stage of the experiment (*p* > 0.05). The results suggest that LP18 can improve egg production in laying hens during the later stage of the experiment while maintaining egg quality. These findings are of great relevance to the poultry industry and may contribute to the production of high-quality eggs.

#### 3.5.2. Egg Quality and Eggshell Ultrastructure

Table 6 presents the noteworthy effects of LP18 on egg quality parameters, including egg shape index, egg albumen height, Haugh unit, and eggshell strength, which increased by 2.53% (*p* <0.05), 18.41% (*p* < 0.01), 12.34% (*p* < 0.01), and 10.28% (*p* < 0.05), respectively, as compared to the CON group. However, LP18 did not exert a significant effect on egg yolk index, yolk color, and eggshell thickness (*p* > 0.05). Scanning electron microscopy was employed to examine the cross-sectional surface and nodules of the eggshell. As shown in Figure 4, LP18 supplementation significantly increased the effective layer thickness, reduced the width, and increased the density of nodules, compared to the CON group. Additionally, scanning electron microscopy revealed that the eggshells of the CON group exhibited numerous wide cracks and large diameter holes, whereas LP18 supplementation significantly decreased the number and width of cracks. These findings provide critical insights into the effects of LP18 supplementation on egg quality, which may have significant implications for the optimization of egg production and the improvement of human dietary intake.

#### 3.5.3. Serum Lipid Metabolism

Supplementation of LP18 in the basal diet of laying hens was found to significantly decrease the concentrations of LDL, FFA, and Gly in the serum of the hens. Specifically, LP18 supplementation led to a 41% reduction in low-density lipoprotein concentrations (*p* < 0.01), and 38% (*p* < 0.05) and 33% (*p* < 0.05) decreases in free fatty acid and glycerin concentrations, respectively. However, LP18 supplementation did not have a significant impact on the concentrations of serum TG or T-CHO (*p* > 0.05) (Table 7). These observations are important as they inform our understanding of the impact of LP18 on lipid metabolism in laying hens, potentially offering novel insights into the regulation of lipid metabolism in other animal systems, including humans.

## 4. Discussion

The global emergence of antimicrobial resistance represents a significant public health concern [25]. Prolonged use of antibiotics disrupts intestinal bacterial communities, leading to alterations in the digestive tract and metabolic processes [26]. Consequently, efforts were directed towards exploring alternatives such as probiotics to mitigate these adverse effects of antibiotics while maintaining or improving production levels [27]. In previous studies, LP18 was identified as the more adhesive strain among the selected bacteria, thus exhibiting excellent potential and safety characteristics as a probiotic [18]. Building upon this knowledge, our research endeavors aimed to elucidate the colonization potential, adhesive properties, and safety implications of LP18 at the genomic level. Subsequently, we evaluated LP18’s efficacy as a dietary supplement during the late phase of egg-laying poultry, focusing specifically on its applicability within the context of poultry production. Our ultimate objective was to holistically assess the capacity of LP18 to enhance both the health and productivity of poultry, thus considering its potential as a valuable feed additive.

The gastrointestinal tract is a complex ecosystem, and the first challenge for probiotics to survive in the gastrointestinal tract is exposure to gastric acid, which can lead to cell inactivation and death due to low pH and high concentrations of pepsin [28]. In our genomic analysis, we identified several genes encoding F0F1-ATPase and sodium proton antiporter/Na-H antiporter, which were also found in *Bacillus velezensis* ZBG17 and are involved in pH regulation [29]. F0F1-ATPase plays a crucial role in expelling H+ from the cell by using ATP, thereby maintaining pH homeostasis and cell viability [30]. Transmembrane Na(+)/H(+) antiporters transport sodium in exchange for H+ across lipid bilayers and are essential for regulating pH balance in the cytoplasm and/or organelles [31]. The existence of *ArgR* proteins, which regulate the biosynthesis of arginine, suggests that LP18 may generate alkaline compounds to neutralize internal pH and adapt to the gastric environment [32].

The second challenge for probiotics is exposure to bile salts in the duodenum, which can alter the lipid composition of the cell membrane and potentially affect cell permeability and the interaction with the membrane environment [28]. Choloylglycine hydrolase, predominantly expressed in *Lactobacillus, Lactococcus, Bacteroides*, and *Pediococcus,* is responsible for the deconjugation (deamidation) of conjugated bile acids [33]. Previous experiments demonstrated the strong acid and bile tolerance of LP18 [18]. In this study, through genomic analysis, we identified genes associated with acid and bile tolerance, providing further validation of the experimental results from a genomic perspective.

Furthermore, we identified several alkaline resistance genes, cold shock proteins (*CspC*, *CspP*, and *CspL*), and chaperone proteins (*DnaK*, *DnaJ*, and *GrpE*) in the LP18 genome. Probiotic growth is sensitive to pH, and alkaline phosphatase controls surface pH through an ATP-dependent mechanism involving bicarbonate secretion [34]. Specifically, *CspL* in the cold shock protein family promotes growth rates at ambient temperatures, enhances cellular thermotolerance at a global transcriptional level, and serves as an mRNA chaperone, regulating global gene expression and appropriately influencing signal transduction pathways under stress conditions [35]. *DnaK* and *GrpE* actively participate in the response to high osmolarity and heat shock by preventing the aggregation of stress-denatured proteins [36]. In *Lactobacillus reuteri* PL503, the *UspA* and *dhaT* genes are associated with regulating MDA-induced oxidative stress [22]. Similarly, *UspA* and *dhaT* genes play crucial roles in antioxidant stress in LP18. ATP-dependent proteases, including *Lon*, *FtsH*, *HslV*, *HslU*, and the *Clp* family, are responsible for intracellular protein degradation in bacteria. These proteases play a vital role in maintaining cellular protein homeostasis by removing damaged, non-functional, and short-lived proteins, particularly under stress conditions that threaten the proteome. Lon, in addition, aids pathogens in evading various forms of stress, including heat, oxidative, and metabolic stress [37]. In summary, from a genomic perspective, LP18 demonstrates the potential for survival in the gastrointestinal tract.

The presence of diverse cell surface proteins in the LP18 genome highlights its multifaceted approach toward gastrointestinal tract colonization. The fibronectin-binding proteins and mucus-binding proteins suggest mechanisms of adhesion to host cells and the mucosal surface, respectively [38]. These findings are supported by the presence of glycosyltransferases and UDP-galactopyranose mutase, which are instrumental in glycosylation processes and bacterial–host interactions [39]. The identified fibronectin-binding proteins (*fbp* genes) further emphasize the colonization potential of LP18, as fibronectin is abundantly present on the host cell surface. Interactions mediated by these adhesions may facilitate the extracellular matrix of mammalian cells and subsequent colonization within the gastrointestinal tract [40]. Sortase (*srtA*) enzymatically cleaves the cell wall sorting motif (LPXTG motif) between threonine and glycine residues, leading to their covalent attachment to the peptidoglycan [41]. The presence of ABC transporters and PTS system proteins, induced by mucin, suggests that LP18 developed strategies to utilize complex sugars and nutrients readily available in the intestinal environment [42]. These transport systems likely support the establishment and survival of LP18 within the host. Overall, these findings provide valuable insights into the genomic repertoire of LP18 and its gastrointestinal tract colonization potential.

LP18, a potential probiotic strain, demonstrates promising characteristics from a genomic perspective, including its ability to survive and adhere within the gastrointestinal tract. Additionally, an analysis of LP18’s safety profile, including antibiotic resistance and virulence factors, further supports its potential for beneficial use. Building upon this genomic analysis, exploring the application of LP18 in the production of laying hens becomes an area of interest. The association with vancomycin resistance was further supported by the discovery of the *vanY* gene in the *vanB* gene cluster in the LP18 genome, which belongs to the glycopeptide resistance gene cluster [43]. In addition, another qacJ resistance gene’s mechanism of resistance involves antibiotic efflux mediated by the small multidrug resistance (SMR) antibiotic efflux pump. The AMR gene family associated with *qacJ* is the SMR antibiotic efflux pump gene family [44]. In the genome of LP18, a total of 48 virulence-associated factors were identified. BLASTP searches in the NCBI gene database revealed the presence of these virulence-related genes in other strains of *L. plantarum*, including strains ATCC 1491 and ZJ316. Therefore, this study concludes that these coding genes are shared among *L. plantarum* strains, and the safety of LP18 is thereby assured. Whole-genome sequencing revealed the presence of two efflux transporters, including a (MATE) efflux transporter, as well as *LmrA* and *LmrB*, which potentially contribute to the inherent antibiotic resistance of the microorganism. This adaptation could enhance its survival within the intestinal microbiota, particularly following antibiotic treatment [45].

In late-stage laying hens, decreased nutrient absorption leads to reduced productive performance, lower egg quality, and subsequent negative economic impacts [46]. Dietary supplementation of probiotics in the basal diets of poultry was shown to enhance intestinal health, thereby contributing to improved productive performance [47], reduced average daily feed intake, and increased feed conversion efficiency [7]. In this study, we observed a decrease in feed conversion efficiency and an increase in egg production rate in laying hens after supplementation with LP18 during the 31–45 day period. The significant impact of LP18 on feed conversion efficiency could potentially be attributed to the enzymatic activity of LP18, which promotes nutrient metabolism and absorption in laying hens [48]. It is possible that LP18 in the gut, in order to produce bacteriocins and volatile antimicrobial substances, may consume excess oxygen, thus increasing nutrient intake and subsequently reducing feed conversion efficiency [46]. Aging laying hens often exhibit decreased egg quality, disrupted intestinal function, and compromised immune response, resulting in significant economic losses. Previous research reports indicated that the addition of probiotics to the diets of breeder hens improves eggshell quality, reduces egg breakage rates, and thereby enhances overall production yield [3]. In this study, we observed significant improvements in the egg shape index, yolk height, Haugh units, and eggshell strength in laying hens fed diets containing LP18. The beneficial impact of probiotics on eggshell quality can be explained by their improvement in calcium availability and absorption. Reports suggest that the inclusion of Bacillus subtilis in the diets of laying ducks contributes to improvements in the egg shape index, Haugh units, and an increasing trend in eggshell strength. Probiotics can enhance eggshell quality by enhancing calcium availability and absorption [49].

An eggshell is composed of a shell membrane, a mammillary layer, and an effective layer consisting of the palisade layer, crystal surface layer, and cuticle [50]. Eggshell strength is largely related to the width of the mammillary knob in the egg ultrastructure [51], as well as the thickness of the effective layer and the density of mammillary knobs [52,53]. The ratio of the palisade layer and mammillary knob density in hens tends to decrease significantly with age [54,55]. The present study revealed that LP18 improves eggshell quality primarily through a reduction in mammillary knob width, as well as a significant increase in effective layer thickness and papillary density. Currently, efforts to improve eggshell ultrastructure largely focused on microelements, such as manganese [56,57], zinc [51,58], copper [59], and chloride [60], with optimal levels and forms found to have positive effects on eggshell quality. Additionally, the supplementation of probiotics was found to enhance intestinal calcium absorption in laying hens during the late production phase, leading to improvements in eggshell quality [61]. LP18 may impact eggshell quality by influencing trace element absorption and increasing the content of inorganic salts.

In the late stage of production, laying hens often exhibit perturbed lipid metabolism characterized by hepatic lipid deposition and abdominal fat accumulation [62]. This dysregulation of lipid metabolism has the potential to profoundly impact the production performance of laying hens [63]. The evaluation of blood lipid levels utilized TG, T-CHO, FFA, LDL, and Gly as markers. Elevated plasma FFA levels result from increased adipose tissue mass, adrenocorticotropic hormone, or other physiological stressors, leading to accelerated adipose tissue lipolysis [64]. FFA promotes lipid droplet accumulation, reduces glycogen synthesis, and upregulates genes involved in lipid synthesis [65]. LDL, a lipoprotein particle, is a form of cholesterol present in the blood and is prone to oxidation into ox-LDL, promoting inflammation [66]. Studies demonstrated that supplementation of probiotics in the maternal chicken diet significantly reduces LDL levels [48]. Furthermore, supplementation of *Clostridium butyricum* was found to accelerate liver fatty acid oxidation, shape the gut microbiota and bile acid profile, and reduce hepatic fat deposition in adult laying hens [67]. Upon LP18 supplementation, while no significant changes were observed in plasma TG and T-CHO concentrations, there was a notable decrease in concentrations of LDL, FFA, and Gly. These findings suggest a potential role of LP18 in regulating lipid metabolism.

## 5. Conclusions

This study sheds light on the genomic characteristics of LP18, revealing traits that enable its survival and colonization in the gastrointestinal tract. It also highlights the potential of LP18 as an effective probiotic for improving productivity, egg quality, and lipid metabolism in poultry production.

## Figures and Tables

**Figure 1 microorganisms-11-02373-f001:**
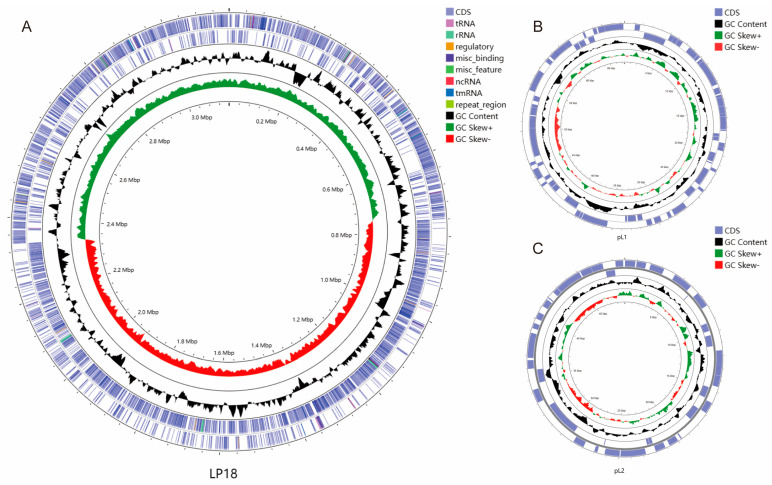
The complete genomic map of LP18. (**A**) LP18 chromosome complete genome. (**B**) Plasmid pL1 complete sequence. (**C**) Plasmid pL2 complete sequence. From inner to outer circles: The first circle depicts the genomic positions. The second circle displays the GC skew values, where the values are plotted as deviations from the average GC skew of the complete sequence, with red indicating values below 0 and green indicating values above 0. The third circle represents the GC content, with regions protruding outward indicating values higher than the average, and regions protruding inward indicating values lower than the average. The fourth circle (forward strand) and the fifth circle (reverse strand) represent the loci of CDS/rRNA/tRNA on the genome, distinguished by different colors (see legend in the upper right corner for reference).

**Figure 2 microorganisms-11-02373-f002:**
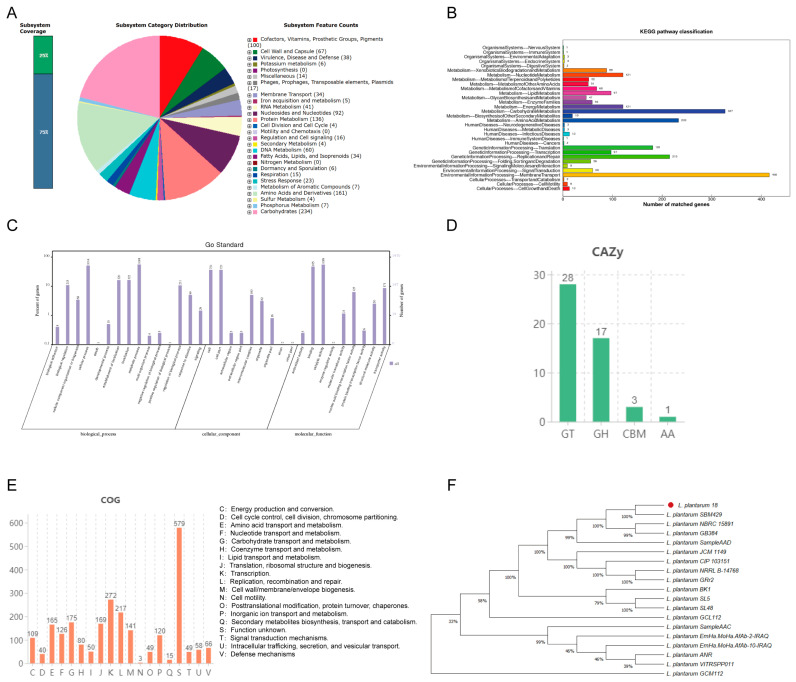
LP18 genome function annotation. (**A**) The classification of RAST annotation in LP18. (**B**) The amino acid sequence of LP18 annotation is based on the KEGG database. (**C**) LP18 annotation based on the GO database. (**D**) LP18 annotation based on the CAZy database. (**E**) LP18 annotation based on the COG database. (**F**) The evolutionary analysis of LP18-based 16S rRNA.

**Figure 3 microorganisms-11-02373-f003:**
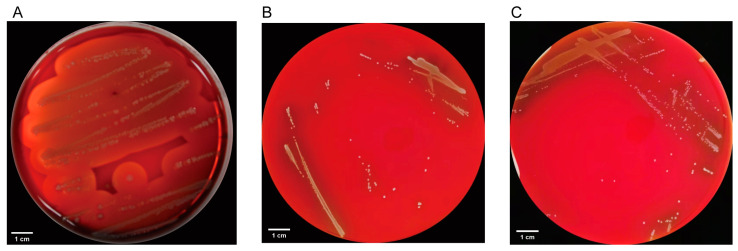
Hemolytic activity of LP18. (**A**) Beta-hemolytic *S. aureus* hemolytic results. (**B**) LP18 hemolytic results. (**C**) Gamma-hemolytic LGG hemolytic results.

**Figure 4 microorganisms-11-02373-f004:**
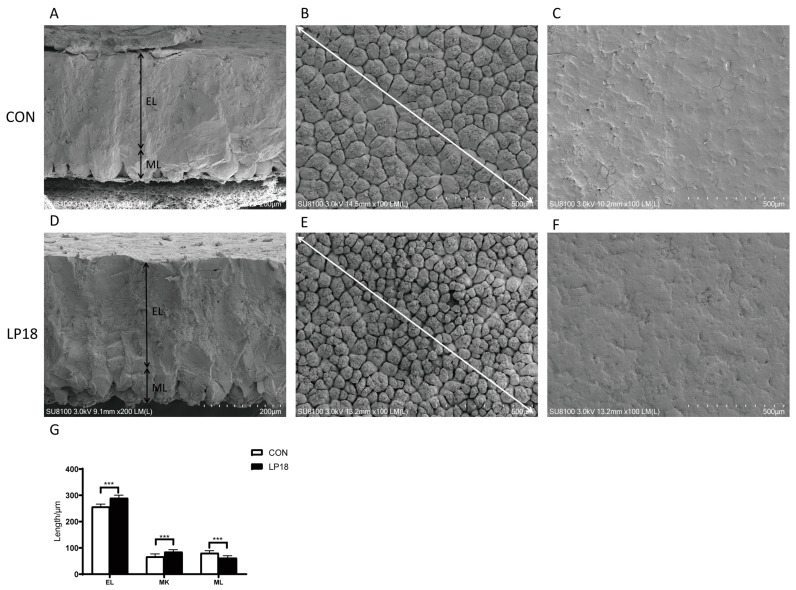
Scanning electron microscopy of the eggshell. Cross-section of the eggshells fed CON (**A**) or LP18 (**D**), scale bar: 200 μm. The inner surface of the eggshells fed CON (**B**) or LP18 (**E**), scale bar: 500 μm. The outer surface of the eggshells fed CON (**C**) or LP18 (**F**), scale bar: 500 μm. (**G**) Eggshell ultrastructure length. EL: Effective layer; ML: mammillary layer; MK: mammillary knots. Data presented indicate the mean ± SEM (*** *p* < 0.001). Arrows in (**B**,**E**): Measurement of the average size of mammillary knot in the scanning electron micrograph. The average size of mammillary knot, s = L/n (where n = the number of mammillary knot, L = intersecting a line of length), was assessed from 2-dimensional pictures.

**Table 1 microorganisms-11-02373-t001:** Genome components of LP18.

Characteristic	Value
Genome size (bp)	3,275,044
GC content (%)	44.42
Gene number	3142
Gene length (bp)	2,732,562
Tandem repeat number	86
Minisatellite DNA number	48
Microsatellite DNA number	4
rRNA number	13
tRNA number	68
sRNA number	3
Genomic island number	4
CRISPR number	1
Prophage number	3

**Table 2 microorganisms-11-02373-t002:** Gastrointestinal tract survival potential related proteins of LP18.

Stress-	Stress-Related Proteins	Query
PH	F0F1-ATPase	CHROMOSOME_1_1226, 1228
	Sodium proton antiporter/Na H antiporter	CHROMOSOME_1_1427, 1476, 1606, 2017, 2394, 2579, 2870, 2948, Plasmid_2_4
	pyruvate kinase	CHROMOSOME_1_764
	*argR*	CHROMOSOME_1_370, 514
	alkaline	CHROMOSOME_1_277, 1413
Bile	*pyrG*	CHROMOSOME_1_2634
	choloylglycine hydrolase	CHROMOSOME_1_1407, 2037, 2160, 2284
Temperature	*csp*	CHROMOSOME_1_42, 175, 2258
	heat shock	CHROMOSOME_1_1472, 2029, 2339
	*dnaJ*	CHROMOSOME_1_870
	*dnaK*	CHROMOSOME_1_871
	*grpE*	CHROMOSOME_1_872
	*hrcA*	CHROMOSOME_1_873
Osmotic stress	ABC transporter	CHROMOSOME_1_117, 319, 335, 411, 775, 820, 1245, 1382, 1431, 1560, 1561, 1677, 1678, 1734, 1740, 1746, 1797, 1913, 2309, 2363, 2417, 2491, 2506, 2516, 2622, 2858
	permease protein	CHROMOSOME_1_318, 2531
Oxidative stress	*UspA*	CHROMOSOME_1_1204, 1455
	*dhaT*	CHROMOSOME_1_1807, 1808
	Glutathione reductase	CHROMOSOME_1_253, 638, 1959, 2553
	*katA*	CHROMOSOME_1_2188
	*Clp*	CHROMOSOME_1_56, 267, 769, 1007, 2194, 2843, 1100
	*Lon*	CHROMOSOME_1_1021
	*Hsl*	CHROMOSOME_1_720, 721

**Table 3 microorganisms-11-02373-t003:** Protein encoding adhesion genes predicted in the genome of LP18.

Protein	Query
Cell surface protein	CHROMOSOME_1_1824, 1743, 1819, 2096, 2224, 1307
WxL domain surface cell wall-binding	CHROMOSOME_1_405, 406, 1052, 1744, 1820, 1825, 2065, 2097, 2098
*FbpA*	CHROMOSOME_1_682, 2410
*srtA*	CHROMOSOME_1_2646
*MucBP*	CHROMOSOME_1_1580, 1411, 2557, 1813, 549, 256, 1337, 2980, 1826, 1
beta-galactosidase	CHROMOSOME_1_2108, 2122, 2122, 2388, 2573
*gap*	CHROMOSOME_1_2846
*glf*	CHROMOSOME_1_222
Glycosyltransferase	CHROMOSOME_1_195, 211, 213, 214, 215, 220, 221, 271, 272, 290, 296, 297, 384, 394, 395, 445, 556, 651, 702, 708, 910, 991, 993, 994, 1028, 1074, 1076, 1336, 1349, 1462, 1507, 1569, 1617, 1618, 2251, 2715, 2463, 2798
PTS_IIA	CHROMOSOME_1_358, 359, 2970, 1566
PTS_IIB	CHROMOSOME_1_360, 361, 1567, 1770, 1933, 2171
PTS_EIIC, PTS_IIB	CHROMOSOME_1_2429

**Table 4 microorganisms-11-02373-t004:** The health status of mice challenged with LP18 for 14 days.

Groups	Deaths Number	Poisoning Symptoms	Anatomical Anomalies	Weight (g)		
				0 d	7 d	14 d
Control	0	NO	NO	22.390 ± 0.211	22.780 ± 0.423	23.210 ± 0.332
LP18	0	NO	NO	22.150 ± 0.434	22.940 ± 0.295	23.330 ± 0.251

**Table 5 microorganisms-11-02373-t005:** Effects of dietary supplementation of LP18 on the production performance of laying hens.

Item	Phase of Experiment	CON	LP18	SEM	*p*-Value
Daily laying rate (%)	1–15 days	72.010	73.330	1.458	0.529
	16–30 days	73.120	73.780	1.693	0.786
	31–45 days	73.320	79.110	1.442	0.009
Feed conversion ratio (Kg feed/Kg egg)	1–15 days	2.220	2.260	0.077	0.762
	16–30 days	2.210	2.120	0.080	0.854
	31–45 days	2.210	2.030	0.053	0.033
Average egg weight (g)	1–15 days	63.780	63.900	0.283	0.761
	16–30 days	64.510	65.080	0.282	0.191
	31–45 days	64.760	64.980	0.299	0.627

**Table 6 microorganisms-11-02373-t006:** The effect of dietary LP18 supplementation on egg quality traits.

Item	CON	LP18	SEM	*p*-Value
Egg shape index	71.890	73.710	0.572	0.032
Yolk height (mm)	5.160	6.110	0.233	0.009
Haugh units	67.410	75.730	1.669	0.002
Yolk index	40.670	41.040	0.399	0.516
Yolk color	7.790	7.920	0.187	0.632
Eggshell strength (N)	33.470	36.910	1.091	0.034
Eggshell thickness (mm)	0.380	0.390	0.007	0.302

**Table 7 microorganisms-11-02373-t007:** Effect of dietary LP18 supplementation on serum lipid parameters of laying hens.

Item	CON	LP18	SEM	*p*-Value
LDL (mmol·L^−1^)	1.720	1.020	0.149	0.005
FFA (mmol·L^−1^)	0.500	0.310	0.047	0.014
Gly (mmol·L^−1^)	65.550	43.870	5.750	0.026
TG (mmol·L^−1^)	21.830	19.350	1.942	0.381
T-CHO (mmol·L^−1^)	3.980	3.640	0.286	0.413

## Data Availability

Genomic data of *Lactiplantibacillus plantarum* 18 has been uploaded to GenBank.

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
