# Peer review of "Genome-Assisted Probiotic Characterization and Application of Lactiplantibacillus plantarum 18 as a Candidate Probiotic for Laying Hen Production"

_microorganisms, 2023, doi:10.3390/microorganisms11102373_

Round 1
Reviewer 1 Report
In this study, the Authors conducted a comprehensive analysis of the Lactiplantibacillus plantarum 18 (LP18) whole genome sequence. Moreover, the efficacy of LP18 on the performance and egg quality of laying hens was evaluated during the late production phase.
Both the idea and hypothesis are good, and the topic investigated fits well the overall scope of the journal.
Moreover, the findings may be useful for poultry production.
The Abstract section needs to be revised by adding some numerical results or significance levels if occurred.
The Introduction section gives a good overview even if the addition of further recently available literature may add value to this section.
The results have been reported in a clear and simple way.
However, the manuscript needs some revision for English language…some sentences result quite hard to follow...
Further, Figures should be provided with more higher quality.
Check if all the references cited into the text have been reported in the references list.
The Conclusions section should be improved because of in its present form is quite hard to follow.
So, after revision the paper could be accepted for publication.
Moderate editing of English language required
Author Response
Dear Reviewer,
Thank you for your valuable feedback and suggestions on our manuscript. We appreciate your positive comments on the overall idea, topic, and findings of our study. We have carefully considered your suggestions and made the necessary revisions to improve the quality of the paper.
We have addressed the reviewers’ comments in a point-by-point manner below, and we have made the necessary revisions to the manuscript. Specific concerns raised by the reviewers have been numbered for clarity. Our responses are given in normal font, and changes/additions to the manuscript are highlighted in red.
Thank you again for your time and consideration.
Response to the comments of Reviewer 1
Point 1:The Abstract section needs to be revised by adding some numerical results or significance levels if occurred.
Response 1: In response to your suggestion, we have added numerical results and significance levels in the revised Abstract section to provide a clearer understanding of our findings.
Point 2:The Introduction section gives a good overview even if the addition of further recently available literature may add value to this section
Response 2: Thank you for your suggestions. We have included recent literature to enhance the Introduction section and provide a more comprehensive overview.
Point 3:However, the manuscript needs some revision for English language…some sentences result quite hard to follow...
Response 3: We feel great thanks for your professional review work on our article. We have taken your comments on the clarity of the language into account and have revised the manuscript to ensure that the sentences are easier to follow and understand.
Point 4:Further, Figures should be provided with more higher quality.
Response 4: We have enhanced the quality of the Figures to provide a better visual representation of our results.
Point 5:Check if all the references cited into the text have been reported in the references list.
Response 5: Thank you for the reminder. We have thoroughly checked the references and made sure that all the citations in the text are included in the references list.
Point 6:The Conclusions section should be improved because of in its present form is quite hard to follow.
Response 6: We feel sorry for our unclear description. We have improved its clarity and coherence to make it easier to follow and understand.
Once again, we sincerely appreciate your time and effort in reviewing our manuscript.
Thank you for your consideration.
Best Wishes to you!
Yours sincerely,
Cuiqing Zhao

Reviewer 2 Report
Dear authors
Thanks for your effort and a nice presentation.
There are few minor comments could be considered during revision.
1. The name of the bacteria should written in a correct way (italic).
2. The bacterial names should be mentioned as full words in the first mention, followed by the abbreviation (in the abstract and the whole manuscript). Then, the name should be mentioned as an abbreviation in the whole manuscript.
3. The name of genes should be mentioned in a correct way.
With my best wishes.
Author Response
Dear Reviewer,
Thank you for your valuable feedback and suggestions on our manuscript. We appreciate your positive comments on the overall idea, topic, and findings of our study. We have carefully considered your suggestions and made the necessary revisions to improve the quality of the paper.
We have addressed the reviewers’ comments in a point-by-point manner below, and we have made the necessary revisions to the manuscript. Specific concerns raised by the reviewers have been numbered for clarity. Our responses are given in normal font, and changes/additions to the manuscript are highlighted in red.
Thank you again for your time and consideration.
Response to the comments of Reviewer 2
Point 1:The name of the bacteria should written in a correct way (italic).
Response 1: Thanks for your correction. We have ensured that the name of the bacteria, Lactiplantibacillus plantarum 18 (LP18), is written correctly in italics throughout the manuscript.
Point 2:The bacterial names should be mentioned as full words in the first mention, followed by the abbreviation (in the abstract and the whole manuscript). Then, the name should be mentioned as an abbreviation in the whole manuscript.
Response 2: We have followed your advice and mentioned the full name of the bacteria followed by the abbreviation in the first mention, both in the Abstract and the rest of the manuscript. After the initial mention, we have used the abbreviation consistently.
Point 3:The name of genes should be mentioned in a correct way.
Response 3: We have reviewed and corrected the gene names to ensure their accuracy and correctness in the revised version of the manuscript.
Once again, we sincerely appreciate your time and effort in reviewing our manuscript.
Thank you for your consideration.
Best Wishes to you!
Yours sincerely,
Cuiqing Zhao

Reviewer 3 Report
This study focuses on the genetic analysis of a specific strain of Lactiplantibacillus plantarum (LP18) to understand its survival and colonization abilities in the gut. The researchers supplemented LP18 in the diet of laying hens and studied its impact on growth, egg quality, and lipid metabolism. The complete genome sequence of LP18 was determined and analyzed, revealing genes related to adaptability, adhesion, and safety in the gut. Supplementation of LP18 resulted in improved productivity, egg quality, and decreased serum lipid content in the hens. These findings demonstrate the potential of LP18 as a probiotic in enhancing poultry production.
Suggestions for improvement:
Line 21, Remove "notable" from the sentence.
Line 41, add full stop after [7]
Remove lines 182-183
Write S. aureus in italics elswhere in the text. Also all the names of bacteria such Clostridium butyricum or others included in the text should be written in italics. Check properly.
I miss the limitation of the study that should be included in the discussion.
Author Response
Dear Reviewer,
Thank you for your valuable feedback and suggestions on our manuscript. We appreciate your positive comments on the overall idea, topic, and findings of our study. We have carefully considered your suggestions and made the necessary revisions to improve the quality of the paper.
We have addressed the reviewers’ comments in a point-by-point manner below, and we have made the necessary revisions to the manuscript. Specific concerns raised by the reviewers have been numbered for clarity. Our responses are given in normal font, and changes/additions to the manuscript are highlighted in red.
Thank you again for your time and consideration.
Response to the comments of Reviewer 3
Point 1:Line 21, Remove "notable" from the sentence.
Line 41, add full stop after [7]
Response 1: Regarding your specific suggestions, we have removed the word "notable" from the mentioned sentence and added a full stop after reference [7] in line 41.
Point 2:Remove lines 182-183
Response 2: We have removed the unnecessary lines 182-183 from the manuscript.
Point 3:Write S. aureus in italics elswhere in the text. Also all the names of bacteria such Clostridium butyricum or others included in the text should be written in italics. Check properly.
Response 3: We have reviewed the manuscript thoroughly to ensure that all bacterial names, including Lactiplantibacillus plantarum, Clostridium butyricum, and others, are written in italics for proper formatting and adherence to scientific conventions.
Once again, we sincerely appreciate your time and effort in reviewing our manuscript.
Thank you for your consideration.
Best Wishes to you!
Yours sincerely,
Cuiqing Zhao

Round 2
Reviewer 1 Report
The revised paper merits the final acceptance.